# Escalating risk-taking is linked to emotional habituation
Hadeel Haj-Ali [1,2] ✉, Moshe Glickman [1,2] & Tali Sharot [1,2,3] ✉

Anecdotally, excessive risk-taking can be traced back to minor acts that escalated gradually. What leads to risk-taking escalation and why is escalation fast in some individuals, but not in others? Here, over three experiments ($N_{\text{Main Experiment}} = 160$, $N_{\text{Validation Experiment}} = 35$, $N_{\text{Control Experiment}} = 30$), we used Virtual Reality to simulate physical risk by having participants walk on a virtual plank suspended in midair. We demonstrate that with repeated opportunities to engage in such risk, emotional responses habituate and risk-taking escalates. The rate of escalation differed dramatically across individuals. We found no credible evidence that individuals' baseline emotions or trait anxiety predicted risk escalation. Instead, the key was how fast anxiety and excitement declined. Individuals who reported faster reduction of anxiety or excitement tended to take more risk over time. The findings may help to identify individuals prone to risk-taking escalation and to develop tools that restore emotions to reduce fatal risk-taking.

Excessive risk-taking can result in dire outcomes, including bankruptcy, physical injury, and death. Many such acts are speculatively traced back to a sequence of smaller instances of risk-taking that gradually escalated. From career-ending political moves to fatal work accidents and sports injuries, commenters retrospectively describe how minor risky decisions snowballed into significant ones over time[1–4]. At the opposite end of the spectrum, extreme risk aversion may result in missed opportunities for personal growth and the avoidance of socially and/or financially beneficial risks[5,6].

Despite the dramatic impact of risk-taking, we do not have a clear understanding of how small risky acts may gradually lead to larger ones, nor why risk-taking escalates more rapidly in some individuals than in others. In fact, most empirical studies examining risk-taking escalation have utilized small financial risks and yielded mixed results[7–9]. Here, we examine whether indeed risk-taking escalates and characterize mechanisms that explain individual differences therein.

When faced with the opportunity to engage in a risky act (e.g., skydiving, gambling large amounts of money), most people will experience negative emotions such as anxiety[10–16]. These emotions serve as cues to either proceed or refrain from engaging in the risky act[17,18]. While strong anxiety will curb risk-taking, the absence of such affective signals will free people to take risks[19,20].

A large body of research demonstrates that the response to an emotion-evoking stimulus weakens with repeated exposure[21–24]. This process, known as emotional habituation, can lead to greater engagement with the emotion-evoking behavior or stimulus[24]. It is thus possible that affective signals that accompany risk-taking also diminish with repetition. If indeed signals that may help curb risk-taking diminish over time, risky acts could increase.

Together with the negative emotions that accompany risk-taking, people may also experience positive anticipatory emotions such as excitement[25–27] that will prompt them to take risks[28]. Excitement is also known to habituate over time with repeated exposure to the eliciting stimulus[29]. In the context of risk-taking, when the excitement is subdued, individuals may decide to take greater and greater risks to recreate the level of excitement they initially experienced[30].

Thus, emotional habituation to both negative and positive feelings may be associated with increased risk-taking. We hypothesize that faster emotional habituation will be linked to quicker risk escalation. This is because (1) as negative emotions (which can restrain risk-taking) diminish, individuals may seek more risk due to reduced emotional barriers, and (2) as positive emotions (like excitement) fade, people may pursue greater risks to recreate those feelings. In both cases, we predict that faster emotional dampening will be related to more rapid risk escalation.

Here, we specifically examine anxiety and excitement as both are anticipatory, high-arousal states that play a role in risk-taking. Anxiety typically arises in contexts of uncertainty or perceived threat[31–33]. Excitement is linked to opportunity, novelty, anticipated reward[27,34–36], and increased risk-seeking[28].

Studying risky decisions in the lab is particularly challenging, as it is unethical to expose participants to high-risk environments that can cause them harm. Even when studying financial risk, participants are usually faced

[1]Affective Brain Lab, Department of Experimental Psychology, University College London, London, UK. [2]The Max Planck UCL Centre for Computational Psychiatry and Ageing Research, University College London, London, UK. [3]Department of Brain and Cognitive Sciences, Massachusetts Institute of Technology, Cambridge, MA, USA. ✉e-mail: hajali.hadeel5@gmail.com; t.sharot@ucl.ac.uk

with low stakes[37,38] that likely do not elicit strong emotional responses[39]. To address this problem, we leverage Virtual Reality (VR) technology. Virtual reality allows realistic simulations of high stakes in a controlled experimental setting[40]. A growing body of evidence demonstrates that VR provides a multi-sensory experience that so accurately simulates real-world scenarios that it evokes genuine behavior and strong affective responses[41–47].

In particular, we used a VR task that simulated a plank suspended at a large height. We quantify risk-taking on a trial-by-trial basis while measuring positive and negative emotional responses. We first tested the validity of the perceived realism of risk by testing for an association between virtual risk-taking with "real-life" fear of heights. In a control experiment, we examined whether risk-taking is independent of motor skill or motor learning. This setup enabled us to investigate whether risk-taking gradually escalates while emotion habituates, and if individual differences in the rate of emotional habituation were linked to the rates of risk escalation.

## Methods

### Participants
Participants were recruited from the UCL SONA recruitment system. Participants with a very significant fear of heights were advised not to sign up for the study to avoid psychological trauma. All subjects provided written consent before the experiment and were paid £6 for their participation. The study was approved by the UCL Psychology Ethics Committee [ID: EP_2023_009]. The sample size was determined based on pilot data ($N = 10$) with an effect size of 0.2 and a power of 80%, with alpha of 0.05 in a linear regression test. Main experiment: $N = 160$ (mean age = 21.8, SD = 4.69; 117 women, 43 men; 104 Asian, 37 White, 10 Other, and 9 African American/Black). Validation experiment: $N = 35$ (mean age = 21.94, SD = 3.69; 25 women, 10 men; 23 Asian, 9 White, and 3 Other); Control experiment: $N = 30$ (mean age = 21.36, SD = 2.68; 25 women, 5 men; 21 Asian, 5 White, 3 Other, and 1 African American/Black). One participant was excluded from the analysis of the main experiment due to VR malfunction. The study was not preregistered.

### Main experiment
**Materials and procedure.** Participants provided written consent and read the task instructions. We used the virtual Richie's Plank Experience, developed by Toast Interactive and obtained from the Meta application store and streamed on a Meta Quest 2 VR headset. The virtual experience simulates a highly realistic and immersive naturalistic environment which includes a virtual plank (Fig. 1a). The experiment room included a physical plank (length = 5.52 m, width = 0.175 m, height = 0.01 m) that aligned with the virtual plank, to make the experience more realistic (Fig. 1a). Participants started each trial (15 in total) on the ground level, inside a virtual elevator. Participants were asked to verbally report their anxiety levels on a scale from 0 (*not at all anxious*) to 10 (*very anxious*) and their excitement level on a scale from 0 (*not at all excited*) to 10 (*very excited*). Participants responded verbally, and the experimenter entered these values into a spreadsheet in real time. An additional experimenter was always present in the room to ensure consistency and accuracy in recording responses.

Next, participants pressed a virtual "Plank" button using the controller, after which the virtual elevator took them up several hundred feet to a virtual plank. The door then opened, and participants waited 5 s until they heard "trial begins." They were then given 20 s to do whatever they wished (e.g., stand still, look around, move, and so on). After 20 s, they were instructed to return to the ground level using a button that brought them back down (Fig. 1a, b). They did so by pressing a button on their controller. This opened a menu where they selected "warp to ground." As soon as they made that selection, they were instantly placed back on the ground. Once they were back on the ground, they were asked to walk back into the elevator. The inter-trial interval was 25 s. Participants were explicitly instructed to press only the "plank" button, as the other buttons were linked to environments not relevant to this study. No participants pressed any of the other buttons.

Participants also reported their demographics, race, education level, experience with VR, general willingness to take risks[48], Trait Anxiety[49], and

were asked, "How afraid are you of heights on a scale from 0 (not at all) to 10 (extremely afraid)?". Participants were asked to state the experiment's aim and completed the Curiosity and Exploration Inventory[50]. The latter was used as a pilot for a future study.

Galvanic Skin Response (GSR) signals were collected at first using the Shimmer GSR+ Unit and the ConsensysPro software (Version V1.6.0). However, the GSR hardware failed after testing 53 participants and so GSR collection was discontinued.

### Analysis
Data preprocessing and analyses were conducted using R[51] within the RStudio environment[52]. Linear mixed-effects models were conducted in RStudio using the afex package[53]. Data visualization was created using the ggplot R package. Bayesian linear regression was performed using JASP[54] (version 0.17.1), with default JZS priors (r-scale = 0.354) to obtain Bayes Factors ($BF_{10}$). We calculated the Bayes Factor for each null effect, which measures the ratio of evidence in favor of the alternative hypothesis compared to the null hypothesis, given the data[55]. A $BF_{10}$ larger than 1 suggests support for the alternative hypothesis and a $BF_{10}$ lower than 1 indicates support for the null hypothesis[56]. For standardization in all analyses reported, we used standard $z$-score normalization (mean = 0, standard deviation = 1) across subjects. All statistical tests were two-sided. Normality of residuals was assessed using the Shapiro–Wilk test and shown to be deviated from normality. Parametric tests were applied under the assumption of normality, supported by the central limit theorem, as all conditions had sufficiently large sample sizes[57].

For linear mixed-effects models, the template for the equation was as follows:

$$\text{Outcome} \sim \text{FixedEffect1} + (1 + \text{FixedEffect1}|\text{Subject}) \qquad (1)$$

**Measures and analysis of risk-taking.** There were two aspects of risk-taking we were interested in. First, we wanted to know how much virtual risk-taking subjects took on average during the task, so we could examine if their virtual risk-taking reflected their reported fear of heights. To that end, we measured risk magnitude per trial, which was defined as the distance walked on the plank in meters (i.e., the farthest point reached on the plank on each trial). This measure ranged from 0 meters (no distance traveled on the plank) to 5.525 meters (the total length of the plank). If participants stepped off the plank they would experience a virtual fall to the ground. Their risk in this case was coded as the value of the maximum risk possible (5.525 meters; a demonstration video of the virtual fall is available at: https://youtube.com/shorts/ZDkBzDFQ7T8?feature=share) (see Supplementary Figs. 1–3 for additional analysis excluding trials where subjects walked off the plank). Using a linear model, we then examined if, across participants, fear of heights was predictive of each participant's average virtual risk-taking magnitude throughout the task.

Once this relationship was validated, we assessed whether virtual risk-taking escalated. In other words—did risk-taking magnitude go up with time across subjects? To test this, we ran a Linear Mixed-Effects Model (LMM) in which risk magnitude (standardized across subjects) is predicted from trial number (with fixed and random slopes and intercepts). We analyzed trials up to and including the one in which the subject took maximum risk (or all trials if the maximum risk was not reached). This would establish if risk escalated.

To quantify each participant's risk escalation rate, we ran individual linear regressions, predicting virtual risk-taking from trial number. The resulting slopes were extracted and standardized across all subjects. This was done to test the correlation between risk-taking escalation and emotional habituation across subjects. Larger slopes indicated a steeper increase in risk-taking, signifying faster risk escalation. Out of 159 participants, eight subjects reached the maximum risk level on the first trial. For these subjects, it was not possible to estimate a risk-taking coefficient over time.

**Fig. 1 | VR task and validation. a** View of the virtual plank and its aligned physical plank in the experiment room. **b** Example trial. Participants started each trial on the ground level inside the elevator, then they were asked to verbally report their anxiety and excitement levels. The elevator then took them up to a plank on which they could walk. On each trial, a participant starts at *point zero* rather than continuing from the point on which they finished the previous trial. **c** A fresh group of participants (*N* = 35) was run to test whether walking further down the plank indeed felt riskier. Upon completing each trial, participants reported their perceived risk on a scale from 0 (*I don't feel this point to be risky at all*) to 10 (*I feel this point to be very risky*). **d** Virtual risk-taking (average distance traveled over all trials for each subject) was negatively associated with the participant's reported fear of heights. Each dot represents a participant. The model prediction is represented by the black line with the shaded areas corresponding to the 95% confidence intervals. ***p* < 0.001.

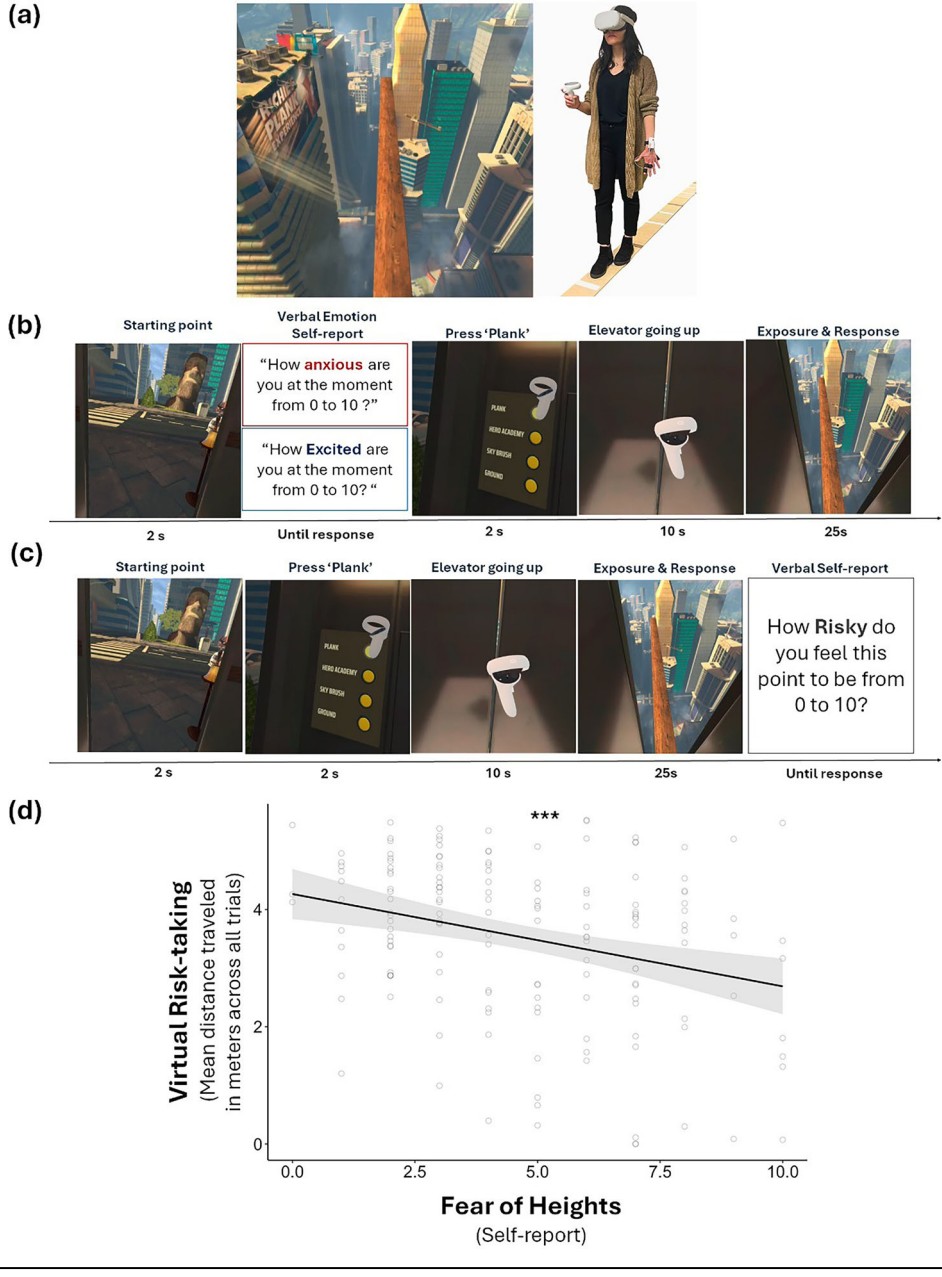

**Measures and analysis of emotional responses.** First, we assessed each participant's *baseline* emotional reaction by collecting self-reported anxiety, and excitement prior to task exposure. Next, we calculated each participant's emotional habituation rates. To that end, for each participant, we ran linear models (one for anxiety, and one for excitement) in which time (defined as trial number) predicts emotional measures from the first task exposure till the last. The slope (standardized across subjects) reflected emotional habituation. *Negative* values indicate habituation (that is a *decrease* in the emotional response over time) while positive values imply sensitization (an increase in the emotional responses over time). Steeper negative slopes, i.e., bigger negative slopes in absolute values, represent greater habituation.

We also tested whether emotional habituation was significant across the sample. In other words—did emotion go down with time across subjects? To test this, we ran Linear Mixed-Effects Models (LMM) (one for anxiety, and one for excitement, standardized across subjects) in which emotion is predicted from trial number (with fixed and random slopes and

intercepts). Again, we expect a *negative* relationship—which would suggest emotional reactions decrease as trial number increases. Four participants showed no change in either anxiety or excitement over time. Thus, it was not possible to estimate their slopes.

**Emotions and risk-taking.** We next examined whether, across participants, emotions were related to risk-taking. Our main aim was to examine if individuals with faster emotional habituation would also show faster risk escalation. To that end, we ran a linear model for each one of the emotion measures gathered (anxiety, excitement, standardized across subjects). This approach allowed us to examine how changes in risk-taking behavior over the course of the experiment were related to each of our measured emotional responses. Finally, to determine whether emotional habituation rates, rather than baseline emotions or trait anxiety levels, predicted the pace of risk escalation, we conducted the same linear models while controlling for baseline emotions and trait anxiety. All scores were standardized across subjects.

### Validation experiment: perceived risk

To examine if walking further down the plank felt riskier to participants, we conducted a validation experiment. A fresh group of participants completed the task as described above, except that at the end of each trial, they verbally provided a report about how risky the point they reached felt to them on a scale from 0 (not at all) to 10 (very risky). They then returned to the ground level by pressing a button that automatically shifted them to the ground level. As the aim of this study was solely to validate individuals' perception of risk, emotional measures (anxiety, excitement) were not collected (Fig. 1c). Participants' demographics including race, and education level were collected. A linear mixed-effect model was run to predict the reported feeling of risk from the distance traveled (fixed and random intercepts and slopes). The main question here was simply—does walking further down the plank feel riskier?—we also had to control for time as a fixed effect in the model because time (i.e. trial number) would likely reduce perceived risk due to habituation.

### Control experiment: motor learning

To answer the question of whether risk escalation and emotional habituation are simply a reflection of motor learning we ran a control task. Here, we again used Riche's plank experience as in the main experiment, except that here the virtual plank was *on the virtual ground level*. The virtual environment included the elevator and a plank that starts at the elevator door, with length and width similar to those used in the main experiment. The plank that participants saw in the virtual environment was the physical plank set up in the room that was presented to them in a virtual mode. Participants verbally provided anxiety and excitement self-reports at the start of each trial. Participants' demographics, race, and education level were collected.

### Reporting summary

Further information on research design is available in the Nature Portfolio Reporting Summary linked to this article.

## Results

We first examined whether, as in real life, participants would perceive walking further and further down the plank in midair as riskier and riskier.

Indeed, they reported feeling they were at greater risk the further they traveled down the plank ($\beta = 0.17$, $t(27.24) = 2.41$, $p = 0.022$, 95% CI [0.027, 0.331]). This confirms that participants' perception of risk increases proportionally with the distance they travel down the plank.

Next, we sought to validate whether individual differences in participants' behavior in the virtual environment represent individual differences in real-world experiences. Specifically, we tested whether self-reported fear of heights predicted virtual risk-taking behavior. As expected, participants who reported greater fear of heights in real life took less virtual risk in the task (linear model predicting virtual risk-taking, defined as the average length walked down the plank, from fear of heights: $\beta = -0.296$, $t(157) = -3.87$, $p < 0.001$, 95% CI [−0.44, −0.14]; Fig. 1d). These findings suggest that the virtual environment effectively elicited behavior consistent with real-world tendencies.

Once we confirmed that the virtual task is indeed experienced as risky and associated with real-world tendencies, we turned to our main research questions: Does risk-taking escalate with repetition? Do associated negative and positive emotions decline with repetition? And is the former linked to the latter?

### Risk-taking escalates with repetition and corresponding emotions habituate

We observed clear evidence that virtual risk-taking escalated, such that participants walked further down the plank as trials progressed (linear mixed-effects model predicting risk-taking from trial number as a fixed effect, with random slopes and intercepts: $\beta = 0.25$, $t(55.56) = 12.36$, $p < 0.001$, 95% CI [0.213, 0.296]; Fig. 2a). Note that for each trial, participants always start at the zero point. They do not accumulate distance across trials; for example, if they walked 3 meters on trial 6, they still started at the zero point on trial 7.

Importantly, across trials, self-reported anxiety decreased as did excitement. In particular, we ran two linear mixed-effects models predicting each emotional response (anxiety, excitement) on each trial from the trial number as a fixed factor with random slopes and intercepts. This analysis revealed that anxiety ($\beta = -0.06$, $t(158.77) = -9.31$, $p < 0.001$, 95% CI [−0.079, −0.051]), and excitement ($\beta = -0.06$, $t(158.31) = -9.59$, $p < 0.001$,

**Fig. 2 | Risk-taking escalates and emotions habituate. a** Virtual risk-taking measured as distance traveled on a trial increased with repetition. **b** Emotional responses decreased with repetition, consistent with habituation, $N = 159$. Bold lines represent the linear model predictions, with shaded areas corresponding to the 95% confidence intervals. Scores were standardized within each subject for visualization purposes. Each dot represents the average Y score across all participants for each trial. Error bars indicate the standard error of the mean. ***$p < 0.001$.

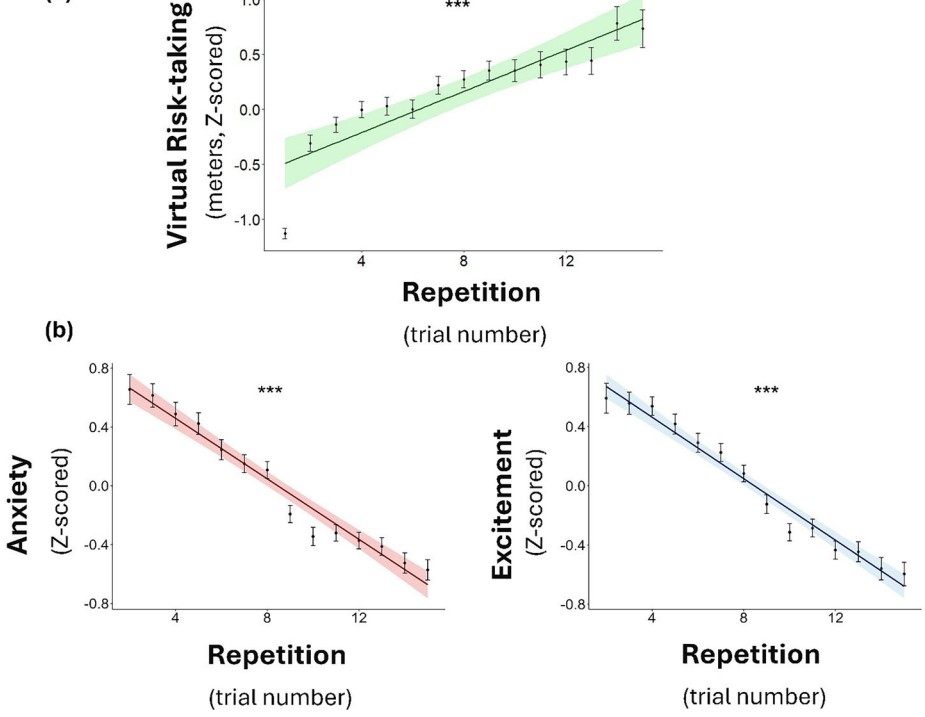

**Fig. 3 | Emotional habituation is associated with faster risk escalation.** Across participants, faster **a** anxiety habituation and **b** excitement habituation were linked to quicker risk escalation, $N = 149$. The habituation rate is calculated as the subject's unstandardized slope in the linear model predicting emotion from trial number ($Z$-scores across subjects). Negative numbers indicate greater habituation, and positive numbers indicate sensitization (i.e., an increase in emotions across trials). Risk escalation rates are calculated as a subject's unstandardized slope in the linear model predicting risk taken from trial number ($Z$-scores across subjects). Positive numbers indicate greater escalation. Each dot represents a subject. Linear model prediction is represented by the bold line with the shaded areas corresponding to the 95% confidence bounds. **$p < 0.01$, *$p < 0.05$.

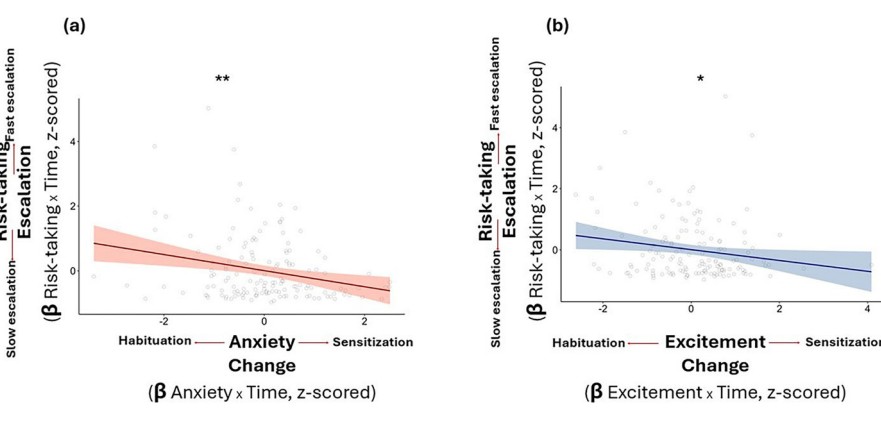

95% CI [−0.084, −0.055]), habituated with repetition. These findings were also true when controlling for distance traveled on each trial by adding it as another fixed variable with random slopes (anxiety: $\beta = −0.06$, $t(152.13) = −8.78$, $p < 0.001$, 95% CI [−0.078, −0.049]; excitement: $\beta = −0.07$, $t(153.87) = −9.95$, $p < 0.001$, 95% CI [−0.086, −0.057]).

There was a small correlation between the average ratings of anxiety and excitement across subjects ($\beta = 0.158$, $t(157) = 2.01$, $p = 0.046$, 95% CI [0.007, 0.318]). That is, subjects who on average gave higher/lower anxiety ratings were slightly more likely to provide higher/lower excitement ratings on average. Moreover, predicting anxiety on a trial-by-trial level from excitement in a linear mixed-effects model with random intercepts and slopes also revealed an association between the two ($\beta = 0.39$, $t(135.208) = 8.98$, $p < 0.001$, 95% CI [0.308, 0.482]). This suggests that both may be partially related to a common feature such as arousal habituation. In addition, across subjects, anxiety and excitement slopes were moderately correlated ($\beta = 0.403$, $t(153) = 5.603$, $p < 0.001$, 95% CI [0.261, 0.546]), indicating that participants who showed a steeper decline in one emotion tended to show a similar decline in the other—consistent with a partially shared process such as arousal habituation. Finally, we fit a linear mixed-effects model predicting trial-by-trial ratings from emotion type (anxiety vs. excitement), trial number, and their interaction. The interaction between emotion and trial was not significant ($\beta = 0.002$, $t(158.305) = 0.505$, $p = 0.614$, 95% CI [−0.005, 0.009]), suggesting that the trajectories of the two emotions did not diverge significantly.

**Faster emotional habituation is associated with faster risk escalation**

Thus far, we have shown that with repeated opportunities to engage in risk, risk-taking escalates while the emotional response to risk habituates. We next examined whether these two phenomena are related.

Indeed, the results indicate that individuals with greater anxiety habituation (defined as a more *negative* relationship between anxiety and trial number) showed the fastest risk escalation ($\beta = −0.249$, $t(146) = −3.14$, $p = 0.002$, 95% CI [−0.406, −0.09]; Fig. 3a).

Moreover, across subjects, those who exhibited a faster pace of excitement habituation (defined as a more *negative* relationship between excitement and trial number) showed faster risk escalation ($\beta = −0.177$, $t(148) = −2.22$, $p = 0.027$, 95% CI [−0.336, −0.019]; Fig. 3b).

The same results are also found when using a different metric of how quickly risk escalates: the number of trials until participants reach maximum risk (i.e., the plank edge). Linear models predicting this measure from each emotional habituation slope (anxiety, excitement) revealed that greater anxiety habituation (more negative slope) correlated with fewer trials to reach maximum risk ($\beta = 0.265$, $t(154) = 3.42$, $p < 0.001$, 95% CI [0.112, 0.419]). Similarly, faster excitement habituation correlated with fewer trials to maximum risk ($\beta = 0.301$, $t(156) = 3.93$, $p < 0.001$, 95% CI [0.149, 0.452]).

Overall, these findings suggest that faster self-reported emotional habituation is associated with risk-taking escalation.

**Emotion habituation, but not baseline emotions nor trait anxiety, predicts risk-taking escalation**

An alternative, though not mutually exclusive account, is that individuals with lower baseline anxiety will show the fastest risk escalation. This, however, was not the case. Entering both baseline anxiety and anxiety habituation into a model that predicts the rate of risk escalation, revealed a significant effect of the rate of anxiety habituation ($\beta = −0.25$, $t(145) = −2.99$, $p = 0.003$, 95% CI [−0.427, −0.087]), but not baseline anxiety ($\beta = −0.02$, $t(145) = −0.24$, $p = 0.809$, 95% CI [−0.191, 0.149], $BF_{10} = 0.04$, which suggests strong evidence for the null hypothesis). In other words, it is not how anxious subjects were before being exposed to the risky environment, but how fast anxiety subsided, which was tied to how quickly they took the ultimate risk in the task.

Similarly, entering both excitement habituation and baseline excitement into a model predicting risk escalation revealed a significant effect of excitement habituation ($\beta = −0.18$, $t(147) = −2.26$, $p = 0.025$, 95% CI [−0.344, −0.023]), but not of baseline excitement ($\beta = −0.04$, $t(147) = −0.48$, $p = 0.627$, 95% CI [−0.203, 0.123], $BF_{10} = 0.12$, which supports the null hypothesis).

Lastly, we assessed whether participants with lower trait anxiety levels (measured using the Trait Anxiety Inventory[49]) showed greater risk escalation. This was not the case—a linear regression revealed that, across subjects, trait anxiety was not associated with the rate of risk escalation ($\beta = 0.13$, $t(149) = 1.59$, $p = 0.114$; 95% CI [−0.031, 0.294], $BF_{10} = 0.55$, which shows anecdotal evidence for the null hypothesis). These findings suggest that neither the intensity of baseline levels of emotions nor trait anxiety are predictive of risk escalation. Instead, the key factor is how emotions evolve with repeated exposure to risk. Note, however, that the Trait Anxiety Inventory may conflate anxiety with depressive symptoms, which could limit its predictive validity[58,59].

Note that in all the analysis above, trials in which subjects stepped off the plank and experienced a virtual fall were scored similarly to walking to the edge, as it is the "riskier" behavior. Just under half of the participants chose to step off the plank at least once during the task, and on average did so 1.8 times. 23.3% of participants did so for the first time within the first third of the experiment, 13.2% during the second third and 9.43% during the final third. Importantly, to ensure that this coding did not impact the results, we reanalyzed the data, excluding all trials in which participants stepped off the plank and found the results unchanged (see Supplementary Figs. 1–3).

Thus far, we have shown that what begins as a relatively minor risky act snowballs into greater risk-taking. At the same time, the associated emotional responses diminish with repeated engagement in risky acts, consistent with habituation. Across participants, faster habituation rates of both

**Fig. 4 | Risk escalation is not due to motor learning.** Displayed on the Y axis is the number of participants who reached the plank's edge on each trial repetition (on the X axis) in the main experiment (N = 159) and the control experiment (N = 30). All but one participant reached the plank edge on their very first attempt in the control experiment, while in the main experiment, only eight subjects did so. This suggests that the gradual escalation in risk-taking is not due to subjects becoming better at walking down a virtual plank.

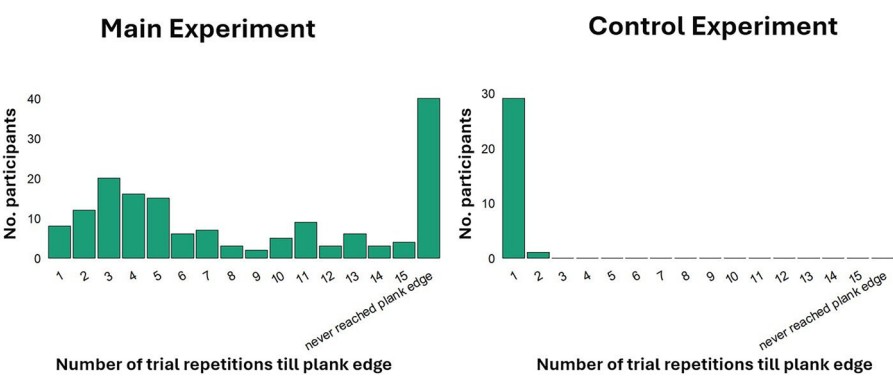

anxiety and excitement were associated with risk escalation. The results underscore the relationship between emotional habituation of both positive and negative feelings and risk escalation.

### Risk escalation is not due to motor learning

To test whether the escalation of risk-taking reported above could be due to an improvement in the participants' ability to walk on a plank with repetition, we conducted a control experiment (N = 30). The experiment was nearly identical to the main experiment described above, except that the virtual plank was shown to participants at ground level. Thus, while walking on the plank requires the same skill as in the main experiment, there was no perceived risk associated with walking on the ground.

As expected, participants reported less anxiety (M = 1.51, SD = 1.41) and excitement (M = 2.96, SD = 1.6) across trials compared to the participants in the main experiment, where perceived risk was present (anxiety: M = 3.08, SD = 1.94, t(187) = −4.21, p < 0.001, Cohen's d = −0.84, 95% CI[−1.24, −0.44]; excitement: M = 5.02, SD = 2.0, t(187) = −5.33, p < 0.001, Cohen's d = −1.06, 95% CI[−1.46, −0.66]).

Most importantly, 29 out of 30 participants reached the edge of the plank on *their first attempt*, with the remaining participant doing so on their second attempt. In contrast, in the main experiment, only 8 out of 159 participants reached the edge of the plank on their *first* attempt. The Kolmogorov–Smirnov test revealed a significant difference between the two distributions (D (159,30) = 0.91, p < 0.001; see Fig. 4), and the means of the two distributions were significantly different as confirmed by the Wilcoxon rank-sum test (W = 130, p < 0.001). These findings suggest that the gradual increase in distance traveled along the plank in the main experiment was not due to gradual learning to walk on a plank.

### Discussion

While taking risks is essential for the progression of individuals and of society, extensive risk-taking can lead to dire consequences. Our results provide empirical evidence for the anecdotal observations that risk-taking escalates with repetition. Within a highly realistic simulated environment, we find that people tend to take relatively minor risks at first, with the magnitude of these risks growing larger over time.

Both the positive and negative self-reported emotions experienced during risk-taking decreased with repetition, consistent with emotional habituation. People reported feeling less and less anxious every time they walked on the virtual high-rise plank, but also less and less excited. Across individuals, those who exhibited the fastest rate of emotional habituation of anxiety and/or excitement levels showed the greatest risk-taking escalation. Neither participants' baseline levels of anxiety and excitement nor their trait anxiety scores predicted the rate of risk escalation. Instead, the only significant predictor of how quickly individuals increased their risk-taking was the rate at which their emotions diminished over time.

While the results are correlational, it is useful to speculate about the potential mechanisms underlying this correlation. In particular, the relationship between emotional habituation and risk-taking may be explained by considering that people rely on emotions to guide their behavior[17], with negative feelings such as anxiety mitigating risk-taking. When those feelings subside because of the general phenomenon of emotional habituation, there will be no signals curbing risk-taking behavior.

Theoretically, positive feelings, such as excitement, may also drive risk-seeking. In fact, in this task, there are no material incentives to take a risk. Thus, people likely decide to walk down the plank because of "internal rewards" associated with curiosity and thrill-seeking. We speculate that as the amount of risk that elicited excitement fails to do so because of emotional habituation, people may be inclined to take larger and larger risks to regain the thrill.

While emotional habituation is well established, its impact on decision-making is not. In fact, an extensive review of the literature[4] has identified only one other study directly measuring the relationship between emotional habituation and decision-making. That study[24] related emotional habituation to dishonesty escalation. The link here, to risk escalation, is not trivial. In many decision-making frameworks, risk preference is treated as a stable trait or as one that is influenced by incidental emotions. In contrast, our study suggests that within-task emotional habituation is related to individual differences in risk-taking. We speculate that emotional habituation may be additionally associated with many other aspects of decision-making.

Our results not only shed light on risk escalation, but also on risk avoidance. We show that reported anxiety is associated with risk avoidance, which in real life can become debilitating. For example, individuals with anxiety disorders may avoid even miniscule risks such as commercial flying. Exposure therapy is often used in the treatment of anxiety disorders to address such issues and is based on the principle of habituation[60]. Through repeated exposure, emotional reactivity to the anxiety-evoking stimulus is reduced, which helps overcome avoidance behaviors. Given the individual differences in emotional habituation we found here, such interventions may benefit from tailoring the number of exposures to each individual's unique rate of habituation.

One strength of the current study is the use of virtual reality technology, which overcomes the limitations of studying risk-taking in traditional laboratory settings[44]. The advantages of using VR in this context are twofold. First, VR allows researchers to expose participants to highly realistic and emotionally engaging scenarios that would be unethical or impractical to recreate in the real world[41–43]. Second, the controlled nature of the virtual environment enables precise measurement and manipulation of variables, facilitating a rigorous experimental design, which would be more difficult to do in the real world. While a virtual environment is not the real world, we did find a negative correlation between the amount of virtual risk the subjects took and their reported fear of heights, providing external validity to the VR design.

## Limitations

Despite strengths of using VR, participants are aware that they are not in actual danger in a VR environment, which may expedite habituation. Another limitation of this study is its correlational nature. While we speculate that a reduction of emotion leads to risk escalation, it is also possible that a third factor explains the correlation. It is less likely, however, that risk escalation would lead to a reduction in anxiety and excitement (rather than an increase). To test for causation, one would need to manipulate the *rate* of emotional habituation directly, which is particularly challenging. Future studies could, however, manipulate the level of emotional arousal (pharmacologically or using relaxation techniques) to different extents along the task to test the effect on risk-taking.

Other future studies could also explore additional relevant traits, such as impulsivity[61], as well as study real-life risk-taking behavior. It would also be interesting to explore emotional habituation in other domains, such as financial risk. Some studies have shown that individuals who engage in more financial risk-taking also seek out more physical risks, such as extreme sports or health-related risks[62–64]. If individual differences in emotional habituation are domain general, they may underlie such associations.

Taken together, the current findings show that with repeated opportunities to engage in risk, risk-taking behaviors escalate and the associated emotions habituate. While emotional habituation is essential for survival by conserving energy and resources[65], it is also associated with increased risk-taking. Knowledge of the role of emotions in the temporal evolution of risk-taking can inform the development of interventions aimed at increasing health and safety. These may include, for example, tools that restore emotional reactions to possible risks on the road or construction sites to reduce fatalities[4].

## Data availability

Data are available online via the lab GitHub at: https://github.com/affective-brain-lab/Risk-Escalation.git.

## Code availability

All codes used to analyze the data are available online via the lab GitHub at: https://github.com/affective-brain-lab/Risk-Escalation.git.

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

## Acknowledgements

H.H.A. is funded by the Economic and Social Research Council (Reference No. ES/P000592/1). T.S. is funded by a Wellcome Trust Senior Research Fellowship (grant no. 214268/Z/18/Z). The funders had no role in study design, data collection and analysis, decision to publish or preparation of the manuscript. We are grateful to Dimitra Klinaki, Kai Tajima, Minjing Chen, Abby Ning, Chen Lu, and Fatema Al-Khalifa for their help in collecting and processing the data. We would also like to thank Noam Markovitch, Rani Moran, Laura Globig, India Pinhorn, Tal Nahari, Jade Serfaty, and Amber Düttmann for discussion and manuscript review.

## Author contributions

H.H.A. was involved in conceptualization, design synthesis, data curation and analysis, data collection, results interpretation, and drafting, editing, and reviewing the manuscript. M.G. was involved in conceptualization, design synthesis, results interpretation, and editing and reviewing the manuscript. T.S. was involved in conceptualization, design synthesis, data curation and analysis, results interpretation, drafting, editing, and reviewing the manuscript, and funding acquisition.

## Competing interests

The authors declare no competing interests.
