## [Transparent Peer Review file · Communications Psychology]

Escalating risk-taking is linked to emotional habituation

Corresponding Author: Ms Hadeel Haj-Ali

Version 0:

Decision Letter:

Dear Ms Haj Ali,

Thank you for your patience during the peer-review process. Your manuscript titled "Feeling Less, Risking More: Emotional Habituation is Linked to Risk-taking Escalation" has now been seen by 3 reviewers, and I include their comments at the end of this message. They find your work of interest but raised some important points. We are interested in the possibility of publishing your study in *Communications Psychology*, but would like to consider your responses to these concerns and assess a revised manuscript before we make a final decision on publication.

We therefore invite you to revise and resubmit your manuscript, along with a point-by-point response to the reviewers. Please highlight all changes in the manuscript text file.

Editorially, we identify three key aspects that must be fully addressed in the revision.

First, a more thorough elaboration of affective constructs is necessary, as all three reviewers have raised concerns in this regard. The revision should clarify the key affective concepts at a theoretical level (e.g., the distinctions between fear, anxiety, and arousal), at an implementational level (e.g., providing a stronger rationale for the choice of dependent variables—why measure anxiety and excitement rather than fear, or why use the STAI instead of impulsivity measures?), and at a statistical level, including supplementary analyses on the temporal relationship between anxiety and excitement.

Second, the concerns raised by Reviewer #3 regarding the correlational nature of the study and its limitations for making causal claims must be fully addressed. Causal claims are only permissible where causal evidence is provided, correlational evidence cannot be used to support causal claims.

Finally, both Reviewer #1 and Reviewer #2 highlight important concerns about the design of the VR experiment.

Specifically, the issue of participants stepping off the plank requires careful examination. If there is no fall animation, participants may learn that stepping off has no consequences, which could introduce a learning effect and serve as a significant confounding variable for the measured effects.

I am attaching an Editorial Requests Table that details critical reporting requirements for the revised manuscript. Please attend to each item and ensure your manuscript is fully compliant. If your revised manuscript is not aligned with these requests on major issues, such as those concerning statistics, it may be returned to you for further revisions without review.

Please submit the following items:

- Revised manuscript
- Point-by-point response to the referees' comments
- Cover letter (as a separate document)

- <https://www.nature.com/documents/nr-reporting-summary.zip>>Nature Research Reporting Summary
- <https://www.nature.com/documents/nr-editorial-policy-checklist.pdf>>Editorial Policy Checklist
- Completed Editorial Request Table (attached).

via this link: Link Redacted .

Additional guidance is available in our style and formatting guide Communications Psychology formatting guide.

Best regards,

Eva R. Pool

Eva R. Pool, PhD
Editorial Board Member
Communications Psychology
orcid.org/0000-0001-5929-1007

REVIEWER EXPERTISE:

Reviewer #1 Emotion and VR
Reviewer #2 Decision making and Risk
Reviewer #3 Modeling and Decision making

REVIEWER REPORTS:

Reviewer #1 (Remarks to the Author):

Dear Dr. Pool,

I read the article written by Ali and colleagues titled "Feeling less, risking more: Emotional habituation is linked to risk-taking escalation".

I found, the article very interesting and the research question is highly relevant not only for the research filed on emotions, but for a broad spectrum of scientific work. I am therefore convinced that this manuscript can be of interest for the readers of the Communication Psychology.

In three independent datasets, the authors investigated why some individuals are prone to behave risky but not others. They used virtual reality (VR) to create a highly controlled and ethically acceptable risky environment (an elevated plank) and they observed the free behavior (i.e., walking on the plank) of the participants. Through repetitions, participants tended to walk longer distance on the plank and this behavior seemed to be guided by a decrease in their anxiety as well as excitement.

I have to say that I do not feel sufficiently expert for evaluating the mathematical calculations applied in the study and therefore I leave the final judgement to other reviewers and the editor. However, while reading the manuscript, I found some unclear details and a few questions raised.

General Comments

1. In the introduction, when describing previous finding in risk taking, the authors refer to fear/anxiety and arousal as two

separate constructs (or at least I understood so). Arousal can be considered a dimension of fear/anxiety responses (e.g., Lang, 1995) and therefore I would not refer to them as if they are two distinct emotional responses.

2. There are a few studies, which already applied height scenario in VR and which I would consider in the introduction as well as in the discussion, even if risk-taking was not directly tested. E.g. Beidermann et al. (2017, BMC Biology); Gromer et al. (2018, Front Hum Neurosci), Gromer et al. (2019, Front Psychol).

3. Anxiety and excitement were considered separately in each analysis. I am in general fine with this approach, but I wondered whether these two emotions were related and whether their decrease throughout the trials was parallel or diverged.

4. Looking at Fig. 4 and reading the discussion, I agree with the authors that participants walked more and more distance, the more repetitions they could have. However, from trial 5 to trial 14, it rather seems that participants preferred to stay safe and not walk much on the plank. Do the authors have any explanation? What does it happen on Trial 15 that all of a sudden so many reached the end of the plank?

Comments about the method

1. What was the rationale for not testing the impulsivity of the participants? Although the State-Trait Anxiety Inventory is one of most commonly used questionnaires for measuring individual anxiety, there is a lot of discussion about its validity as several items can be more valid for depressive mood rather than anxiety.

2. From Fig. 1, I can see four buttons in the elevator. Did participants try to press the other buttons? If yes, what did it then happen?

3. At the end of a trial, participants “were instructed to return to the ground level using a button that brought them back down” (Lines 369-370). How exactly did this work? Was the instruction played as the one for the trial begin? Did the participants had time to return to the elevator once the instruction was played? If yes, how much time? Did the participants had to actively press the button or did the elevator start automatically to descent?

4. How exactly were the ratings collected?

5. Apparently, a few participants tested the virtual environment and walked outside the plank. I think, this is a special behavior, but the authors simply coded those trials as if participants walked to the edge of the plank. How many participants and respectively how many times did individuals step outside? Was this behavior performed at the very beginning of the experimental session or rather at the end?

6. Can the authors better specify what the 0 meter was? Was it the start of the plank from the elevator or was it the starting position in the elevator? I think, this might be related to the negative scores in Fig2a. If not, what are these negative scores?

7. To me, it remained unclear how fear of heights was measured. Did the authors collect a questionnaire or simply asked?

8. What was the rationale for not considering trial in the LMM of the first sample (real life risk vs. virtual risk)?

Minor comments

1. I would briefly describe the protocol in the abstract.

2. The authors should consider this article as well: Andreatta et al. (2023, Springer Berlin).

3. Please add whether the ethic approval was in accordance with the declaration of Helsinki.

4. How high was the plank from the ground?

5. Please define each abbreviation (e.g., GSR).

6. Please specify in the method the standardization of the ratings. In the graphs, I see that the authors used z-scores, but this need to be indicated in the method.

Reviewer #2 (Remarks to the Author):

This manuscript describes an experiment that makes use of virtual reality to simulate walking on a plank high above the ground. It finds that after repeated trials, decreases in ratings of anxiety and excitement are correlated with increased simulated risk-taking (participants walk farther across the virtual plank). This is an original study that cleverly makes use of VR technology to simulate risk. However, there are some drawbacks to this approach, since participants are cognitively aware there is no actual risk. Additionally, the manuscript would benefit from more discussion of the potential benefits of taking on some amount of risk, and the potential implications of the study for anxiety disorders. The study will likely be of interest to researchers studying risk and decision-making. More detailed comments are listed below:

-The manuscript frames risk-taking as dangerous and something to be minimized. However, a lack of risk-taking can also be detrimental. I think the manuscript would benefit from reframing it around the need for a balance between risk-taking and risk-avoidance. As the results from the reported experiment show, high anxiety is associated with the avoidance of risk. People with pathological levels of anxiety can engage in avoidance behaviors that keep them from taking beneficial risks. Therefore, the results have implications not just for people who engage in excessive risk-taking, but also for people with anxiety disorders. Habituation to anxiety-provoking risks could be beneficial for such people (as demonstrated by exposure therapy). More discussion of the positive aspects of taking on risk and the therapeutic implications for anxiety is warranted.

-The use of virtual reality is a clever way to simulate risk. However, within the experiment, participants cognitively know they are safe, even though they feel emotions related to physical risk, such as anxiety and excitement. The lack of actual risk may have increased the habituation of anxiety and excitement within the experiment, relative to a real-world scenario where pain or injury could occur. Discussion of the limitations of using VR for the experiment is warranted.

-Relatedly, more information is needed about what happens when participants fall within the experiment. How often did this occur? Were there ratings of fear or anxiety experienced during the falls? What happened upon virtual “impact” with the ground?

-How were anxiety and excitement selected as the emotions to include in the experiment? What distinguishes anxiety and fear in this experiment? Were any other emotions measured? Since the study was not preregistered, it is hard to tell why certain experimental decisions were made, such as which measures to use and which analyses to run.

-The discussion mainly restates the results. Emphasizing the theoretical contribution, the practical implications, and a more thorough discussion of limitations would be useful.

Reviewer #3 (Remarks to the Author):

Ali and colleagues examined the affective dynamics during risk-taking escalation in a VR setting, and have found that (1) emotion habituates; (2) risk taking escalates; and (3) the rate of emotion habituation correlates with the rate of risk escalation. I have reviewed an RLDM submission related to this study, so some of the comments below maybe related.

While the main results of the experiment are interesting, my biggest concern for the paper is the correlational nature of the findings on emotion habituation and risk escalation. Though the authors acknowledged this in the discussion, throughout the paper (including the introduction), the authors hint at a causal relationship between these two variables. E.g., in the abstract: 'the reduction of excitement may have driven individuals to take more risks to reinstate such feelings'. In the introduction: 'Thus, emotional habituation to both negative and positive feelings may lead to increased risk-taking'. Given the current dataset, I am trying to imagine how the authors could test this - the first thing that comes to mind is to use the baseline emotion (before the experiment started) and other trait measures and see if they predict risk-taking, which the authors have done and found no relationship. Therefore, I would strongly suggest the authors tone down their language.

Another thing that I am confused about in the paper is the fact that people keep taking risks and that their anxiety seems to be reducing. If as the authors have argued a lower level of positive/negative emotion promotes more risk taking & if we assume people have an optimal level of anxiety/excitement, would they eventually converge to a stable level of emotion but taking more and more risk (since emotion itself will get habituate as time goes by?) Do the authors agree with the above argument but just the current task is not long enough for them to converge to that saturation point?

If the authors ditched the hint at causality, then I am not sure that this paper has added too much to what we already know (given the current dataset), since as the authors have mentioned, emotion habituates in response to an emotion-evoking stimulus is an established finding. Extra experiments can potentially make the current study interesting e.g., as the authors have alluded to, manipulate emotion separately on a trial-by-trial basis (so that the authors can manipulate the rate of the outcome of emotion habituation) and see if it causally influences risk taking behavior. Another way could be using a yoked design and making limitations/requirements on the amount of risk the participant needs to take, and comparing the emotion habituation level between (1) voluntarily taking however amount of risk and (2) pre-specified high/low risk level.

Other comments:

Though the authors have argued for the link between the in-lab measure and real-life counterpart (afraid of heights), I am not sure that this can be directly linked to risk escalation in economic domains - is there any work suggesting that physical risk-taking correlates with financial risk taking? The author should discuss this point in the discussion.

In the introduction, the authors argue that 'while strong fear and anxiety will curb risk-taking, the absence of such affective signals will free people to take risks' - this is not necessarily true. In FeldmanHall et al. (2016), they found that this depends on whether the situation is risk or ambiguous. Under ambiguity, higher physiological arousal predicts more ambiguous seeking behavior. The concept of risk in their paper does not necessarily map onto the current paper - if I have to choose, I would say that risk taking in the current experiment is more related to ambiguity than risk since the probability of falling down is unknown to the participant. Relatedly, RE the relationship between physiological arousal and risk-taking, this seems can be tested by some physiological data, which the authors seem to have, at least for a subset of all the participants (i.e., GSR data for the first 53 participants). I am not sure if the data will be underpowered to test this, but it would be interesting to see this set of results.

In the methods, the authors mentioned that if the participant fell off, it is calculated as the maximum length - why is this the case? what is the proportion of the data that belongs to falling off?

Is there any correlation between anxiety and excitement? Do the authors believe that anxiety and excitement have a common cause, or do they originate separately?

The second paragraph mentions Figure 1c as the graph showing some coefficients - I am not sure how a graphical illustration of the experiment suffices this goal - maybe the authors mean a different graph?

* TRANSPARENT PEER REVIEW: Communications Psychology uses a transparent peer review system. This means that we publish the editorial decision letters including Reviewers' comments to the authors and the author rebuttal letters online as a supplementary peer review file. However, on author request, confidential information and data can be removed from the published reviewer reports and rebuttal letters prior to publication. If your manuscript has been previously reviewed at another journal, those Reviewers' comments would not form part of the published peer review file.

If you experience problems in linking your ORCID, please contact the Platform Support Helpdesk.

Version 1:

Decision Letter:

Dear Ms Haj-Ali,

Your manuscript titled "Feeling Less, Risking More: Emotional Habituation is Linked to Risk-taking Escalation" has now been seen by our reviewers, whose comments appear below. In light of their advice I am delighted to say that we are happy, in principle, to publish a suitably revised version in Communications Psychology.

We therefore invite you to revise your paper one last time to address the remaining concerns of our reviewers and a list of editorial requests. At the same time we ask that you edit your manuscript to comply with our format requirements and to maximise the accessibility and therefore the impact of your work.

EDITORIAL REQUESTS:

SUBMISSION INFORMATION:

OPEN ACCESS:

* TRANSPARENT PEER REVIEW: Communications Psychology uses a transparent peer review system. On author request, confidential information and data can be removed from the published reviewer reports and rebuttal letters prior to

publication. If you are concerned about the release of confidential data, please let us know specifically what information you would like to have removed. Please note that we cannot incorporate redactions for any other reasons.

* **DATA AVAILABILITY:**

Link Redacted

Best regards,

Jennifer Bellingtier

Jennifer Bellingtier, PhD
Senior Editor
Communications Psychology

Eva R. Pool, PhD
Editorial Board Member
Communications Psychology
orcid.org/0000-0001-5929-1007

REVIEWER EXPERTISE:

Reviewer #1 Emotion and VR
Reviewer #2 Decision making and Risk

REVIEWERS' COMMENTS:

Reviewer #1 (Remarks to the Author):

The authors responded to all my questions and I found the article well improved.
I do not have any further comment.

Reviewer #2 (Remarks to the Author):

The authors have been responsive to the reviews, and the manuscript has improved as a result of the revisions. I have no further suggestions.

We are grateful to the reviewers for their thoughtful and constructive suggestions, which have helped us improve our manuscript. Below you will find a detailed, point-by-point response, with reviewer comments in bold followed by our responses.

We hope you will now find the manuscript ready for publication in *Communication Psychology*.

Thank you for your consideration,

Hadeel Haj-Ali, Moshe Glickman, and Tali Sharot

University College London & Massachusetts Institute of Technology

Reviewer #1

I read the article written by Ali and colleagues titled “Feeling less, risking more: Emotional habituation is linked to risk-taking escalation”. I found, the article very interesting and the research question is highly relevant not only for the research filed on emotions, but for a broad spectrum of scientific work. I am therefore convinced that this manuscript can be of interest for the readers of the *Communication Psychology*. In three independent datasets, the authors investigated why some individuals are prone to behave risky but not others. They used virtual reality (VR) to create a highly controlled and ethically acceptable risky environment (an elevated plank) and they observed the free behavior (i.e., walking on the plank) of the participants. Through repetitions, participants tended to walk longer distance on the plank and this behavior seemed to be guided by a decrease in their anxiety as well as excitement.

- We thank the reviewer for this positive assessment.

I have to say that I do not feel sufficiently expert for evaluating the mathematical calculations applied in the study and therefore I leave the final judgement to other reviewers and the editor. However, while reading the manuscript, I found some unclear details and a few questions raised.

General Comments

1- In the introduction, when describing previous finding in risk taking, the authors refer to fear/anxiety and arousal as two separate constructs (or at least I understood so). Arousal can be considered a dimension of fear/anxiety responses (e.g., Lang, 1995) and therefore I would not refer to them as if they are two distinct emotional responses.

- We thank the reviewer for pointing this out. Indeed, arousal is a core dimension of emotional responses (Lang, 1995; Russell & Barrett, 1999). We have now

reworded the introduction to avoid the impression that we are referring to arousal as a distinct response. (p. 3).

2- There are a few studies, which already applied height scenario in VR and which I would consider in the introduction as well as in the discussion, even if risk-taking was not directly tested. E.g. Beidermann et al. (2017, BMC Biology); Gromer et al. (2018, Front Hum Neurosci), Gromer et al. (2019, Front Psychol).

➤ We thank the reviewer for drawing our attention to these valuable studies which further validate virtual reality as a tool to elicit emotional responses. We now cite Biedermann et al. (2017), Gromer et al. (2018), and Gromer et al. (2019) in the introduction and Discussion sections (pp. 4, 15).

3- Anxiety and excitement were considered separately in each analysis. I am in general fine with this approach, but I wondered whether these two emotions were related and whether their decrease throughout the trials was parallel or diverged.

➤ There was a small correlation between the average ratings of anxiety and excitement across subjects ($\beta = 0.158$, $t(157) = 2.01$, $P = 0.046$). That is, subjects who on average gave higher/lower anxiety ratings were slightly more likely to provide higher/lower excitement ratings on average. Moreover, predicting anxiety on a trial-by-trial level from excitement in a mixed linear model with random intercepts and slopes also revealed an association between the two ($\beta = 0.39$, $t(135.208) = 8.98$, $P < 0.001$). This suggests that both may be partially related to a common feature such as arousal. In addition, subjects' anxiety and excitement slopes were moderately correlated ($\beta = 0.403$, $t(153) = 5.603$, $P < 0.001$) indicating that participants who showed a steeper decline in one emotion tended to show a similar decline in the other — consistent with a partially shared process such as arousal habituation. Finally, we fit a linear mixed-effects model predicting trial-by-trial ratings from emotion type (anxiety vs. excitement), trial number, and their interaction. The interaction between emotion and trial was not significant ($\beta = 0.002$, $t(158.305) = 0.505$, $P = 0.614$), suggesting that the trajectories of these two emotions do not diverge significantly. These additional analyses have been included in the Results section (p. 11).

4- Looking at Fig. 4 and reading the discussion, I agree with the authors that participants walked more and more distance, the more repetitions they could have. However, from trial 5 to trial 14, it rather seems that participants preferred to stay safe and not walk much on the plank. Do the authors have any explanation? What does it happen on Trial 15 that all of a sudden so many reached the end of the plank?

- The reviewer's comment highlighted that the figure was confusing. The bar labeled '15' included subjects that reached the edge on the 15th trial and those who never reached the edge at all. In fact, most of those included in that bar never reached the edge. This information was in the figure caption but clearly was not the optimal way to present the data. We have now revised the x axis to separate those that reached the edge on the 15th trial and those that never reached the edge (p. 14).

Methodology

5- What was the rationale for not testing the impulsivity of the participants? Although the State-Trait Anxiety Inventory is one of most commonly used questionnaires for measuring individual anxiety, there is a lot of discussion about its validity as several items can be more valid for depressive mood rather than anxiety.

- We measured trait anxiety to answer the question of whether it was anxiety habituation and/or trait anxiety that explain risk-taking escalation. We thank the reviewer for highlighting the discussion around STAI which we now cite (e.g., Bieling, Antony, & Swinson, 1998) and interpreted the results on trait anxiety cautiously (pp. 12-13). We did not measure impulsivity as we had no *a priori* hypothesis about its role in emotional habituation and risk escalation. However, we now mention that it would be interesting to study the relationship between impulsivity and habituation and risk escalation in future studies (pp. 15-16).

6- From Fig. 1, I can see four buttons in the elevator. Did participants try to press the other buttons? If yes, what did it then happen?

- Participants were explicitly instructed to press only the 'plank' button, as the other buttons were linked to environments not relevant to this study. No participants pressed any of the other buttons. We have now added this clarification to the Methods section (p. 5).

7- At the end of a trial, participants "were instructed to return to the ground level using a button that brought them back down" (Lines 369-370). How exactly did this work? Was the instruction played as the one for the trial begin? Did the participants had time to return to the elevator once the instruction was played? If yes, how much time? Did the participants had to actively press the button or did the elevator start automatically to descent?

We have now added a clearer explanation to the Methods section (pp. 4-5). At the end of each trial, the experimenter instructed participants verbally to press a

button on their controller. This opened a menu where they selected “warp to ground.” As soon as they made that selection, they were instantly placed back on the ground. Once they were back on the ground, they were asked to walk back into the elevator. The next inter trial interval (25 seconds) started as soon as they returned to the elevator.

8- How exactly were the ratings collected?

- Ratings were collected verbally. Specifically, participants were asked at the start of each trial, “How anxious are you right now, from 0 to 10?” followed by, “How excited are you right now, from 0 to 10?”. Participants responded verbally, and the experimenter entered these values into a spreadsheet in real time. An additional experimenter was always present in the room to ensure consistency and accuracy. We have now added this clarification to the Methods section (pp. 4-5).

9- Apparently, a few participants tested the virtual environment and walked outside the plank. I think, this is a special behavior, but the authors simply coded those trials as if participants walked to the edge of the plank. How many participants and respectively how many times did individuals step outside? Was this behavior performed at the very beginning of the experimental session or rather at the end?

- Just under half of the participants chose to step off the plank at least once during the task. 23.3% did so for the first time within the first third of the experiment (trials 1–5), 13.2% during the second third and 9.43% during the final third (trials 11–15). We gave these trials the same score as walking to the edge as it is the ‘riskier’ behavior, conceptually equivalent to walking to the end of the plank. However, to ensure that this coding does not impact the results we reanalyzed the data excluding all trials in which participants stepped off the plank and find the results unchanged (see Supplementary Results pp.1-2).

10- Can the authors better specify what the 0 meter was? Was it the start of the plank from the elevator or was it the starting position in the elevator? I think, this might be related to the negative scores in Fig2a. If not, what are these negative scores?

- A zero indicates that the subject did not step onto the plank from the elevator. In Figure 2a, the values on the y-axis reflect **standardized** (Z-scored) risk-taking values. The negative values are negative Z scores (not negative raw numbers), corresponding to below-average risk-taking relative to the sample. We have now clarified this further in the figure caption (pp. 10-11).

11- To me, it remained unclear how fear of heights was measured. Did the authors collect a questionnaire or simply asked?

- Fear of heights was measured by asking participants were asked, "How afraid are you of heights on a scale from 0 (not at all) to 10 (extremely afraid)?" We have now added this information to the Methods section (p. 5)

12- What was the rationale for not considering trial in the LMM of the first sample (real life risk vs. virtual risk)?

- This analysis examines for a correlation across individuals, asking whether people who report taking more risk in real life also take more virtual risk. It looks at between-subject differences rather than within-subject trial-level variation.

Minor comments

13- I would briefly describe the protocol in the abstract.

- We have now added a brief description of the protocol to the abstract while maintain the word limit allowed (p. 2).

14- The authors should consider this article as well: Andreatta et al. (2023, Springer Berlin).

- Thank you for pointing us to this important study on VR which we now have referenced in both the Introduction and Discussion sections (pp. 4, 15).

15- Please add whether the ethic approval was in accordance with the declaration of Helsinki.

- The study was approved by the UCL Experimental Psychology Ethics Committee (ID: EP_2023_009) and was conducted in accordance with strict ethical guidelines consistent with the principles outlined in the Declaration of Helsinki. It met all governance and ethical standards for research involving human participants. We now mention this in the Method section (p. 5).

16- How high was the plank from the ground?

- The actual (real) plank was 0.01 m off the ground. This information has now been added to the Methods section (p. 5).

17- Please define each abbreviation (e.g., GSR).

- GSR stands for Galvanic Skin Response. We have now clarified this abbreviation in the manuscript (p. 5).

18- Please specify in the method the standardization of the ratings. In the graphs, I see that the authors used z-scores, but this need to be indicated in the method.

- We used standard z-score normalization (mean = 0, standard deviation = 1) across subjects. This information has now been added to the Methods section (p. 5).

Reviewer #2:

This manuscript describes an experiment that makes use of virtual reality to simulate walking on a plank high above the ground. It finds that after repeated trials, decreases in ratings of anxiety and excitement are correlated with increased simulated risk-taking (participants walk farther across the virtual plank). This is an original study that cleverly makes use of VR technology to simulate risk.

- We thank the reviewer for their positive feedback.

However, there are some drawbacks to this approach, since participants are cognitively aware there is no actual risk. Additionally, the manuscript would benefit from more discussion of the potential benefits of taking on some amount of risk, and the potential implications of the study for anxiety disorders. The study will likely be of interest to researchers studying risk and decision-making. More detailed comments are listed below:

1- The manuscript frames risk-taking as dangerous and something to be minimized. However, a lack of risk-taking can also be detrimental. I think the manuscript would benefit from reframing it around the need for a balance between risk-taking and risk-avoidance. As the results from the reported experiment show, high anxiety is associated with the avoidance of risk. People with pathological levels of anxiety can engage in avoidance behaviors that keep them from taking beneficial risks. Therefore, the results have implications not just for people who engage in excessive risk-taking, but also for people with anxiety disorders. Habituation to anxiety-provoking risks could be beneficial for such people (as demonstrated by exposure therapy). More discussion of the positive aspects of taking on risk and the therapeutic implications for anxiety is warranted.

- The reviewer is absolutely right. We have now revised both the Introduction and Discussion to highlight also the importance of risk taking for optimal behavior and the downsides to compulsively avoiding risk (pp. 3-4, 15).

2- The use of virtual reality is a clever way to simulate risk. However, within the experiment, participants cognitively know they are safe, even though they feel emotions related to physical risk, such as anxiety and excitement. The lack of actual risk may have increased the habituation of anxiety and excitement within the experiment, relative to a real-world scenario where pain or injury could occur. Discussion of the limitations of using VR for the experiment is warranted.

- Following the reviewers' suggestion we have now added a discussion of the limitations of using VR to the Discussion section (pp. 15-16) including the possibility that habituation might occur faster in VR.

3- Relatedly, more information is needed about what happens when participants fall within the experiment. How often did this occur? Were there ratings of fear or anxiety experienced during the falls? What happened upon virtual "impact" with the ground?

- In trials where participants chose to step off the plank entirely, they experienced a virtual fall to the ground. Upon impact, the screen momentarily turns white, and a few seconds later the environment transitions to the ground-level view. We have now added a video of the virtual fall sequence to the Methods section (p. 6). Just under half of the participants chose to step off the plank at least once during the task (Supplementary Results pp. 1-2). Because self-report measures were always collected at the start of each trial (before taking the elevator to the plank) ratings would not have been collected during falls. To ensure that the main findings were not driven by fall-related trials, we reanalyzed the data excluding all trials where participants stepped off the plank - the results stay the same (Supplementary Results pp. 1-2).

4- How were anxiety and excitement selected as the emotions to include in the experiment?

- We thank the reviewer for prompting us to clarify our selection of emotional measures. We selected anxiety and excitement as they are both anticipatory states of heightened arousal that are known to modulate decision-making under risk. Anxiety arises in contexts of uncertainty or potential threat (Gray, 1991; Beck, Emery, & Greenberg, 2005; Grupe & Nitschke, 2013) and drives risk-averse behavior (Kuhnen & Knutson, 2011; Miu, Heilman, & Houser, 2008). Excitement is linked to opportunity, novelty, or anticipated reward (Kuppens et al., 2013; Brooks, 2014; Ketonen et al., 2023) and increased risk-seeking behaviours (e.g., Kuhnen & Knutson, 2011). We now add the above to the Introduction section (p. 4).

5- What distinguishes anxiety and fear in this experiment?

- Anxiety is usually thought of as an anticipatory emotion and fear as an emotional reaction to one's immediate environment (Davis et al., 2010; Daniel-Watanabe & Fletcher, 2021; Gray, 1991; Beck, Emery, & Greenberg, 2005). We were interested in anticipatory emotions and thus measures anxiety (and excitement) when the subjects were in the elevator, before stepping on the plank.

6- Were any other emotions measured?

- No other emotions were measured

7- Since the study was not preregistered, it is hard to tell why certain experimental decisions were made, such as which measures to use and which analyses to run.

- We thank the reviewer for highlighting the importance of clarifying our rationale for key design and analytical decisions. We selected anxiety and excitement as our primary emotional measures because they are both anticipatory, high-arousal states that are widely recognized as central to risk-related decision-making. Anxiety arises in situations of uncertainty or perceived threat (Gray, 1991; Beck, Emery, & Greenberg, 2005; Grupe & Nitschke, 2013) and has been consistently shown to drive risk-averse behavior (e.g., Kuhnen & Knutson, 2011; Miu, Heilman, & Houser, 2008). Excitement, in contrast, is associated with opportunity, novelty, and anticipated reward (Kuppens et al., 2013; Brooks, 2014; Ketonen et al., 2023) and has been linked to increased risk-seeking behavior (e.g., Kuhnen & Knutson, 2011).

We operationalized risk-taking as the furthest distance walked on the plank. To validate this measure, we conducted a separate validation experiment (pp. 7-8) showing that the distance participants walked strongly correlated with self-reported risk perception—i.e., greater distances were perceived as riskier.

Our main hypothesis is centered on whether the rate of emotional habituation would predict the rate of risk escalation across individuals. To directly test this hypothesis, we calculated slopes for each participant's trial-by-trial emotional responses and risk-taking behavior. These individual-level slopes served as quantitative indices of habituation and escalation, respectively, allowing us to conduct a one-to-one analysis of their relationship.

8- The discussion mainly restates the results. Emphasizing the theoretical contribution, the practical implications, and a more thorough discussion of limitations would be useful.

- This is a good suggestion. Following the reviewer's comments, we have revised the Discussion section to better highlight the broader relevance of both excessive risk-seeking and risk avoidance in theoretical and practical contexts. We now elaborate on how these findings contribute to the literature regarding risk behavior. Additionally, we discuss the limitations and possible future research directions (pp. 14-16).

Reviewer #3:

Ali and colleagues examined the affective dynamics during risk-taking escalation in a VR setting, and have found that (1) emotion habituates; (2) risk taking escalates; and (3) the rate of emotion habituation correlates with the rate of risk escalation. I have reviewed an RLDM submission related to this study, so some of the comments below maybe related.

Major Comments:

- 1- While the main results of the experiment are interesting, my biggest concern for the paper is the correlational nature of the findings on emotion habituation and risk escalation. Though the authors acknowledged this in the discussion, throughout the paper (including the introduction), the authors hint at a causal relationship between these two variables. E.g., in the abstract: 'the reduction of excitement may have driven individuals to take more risks to reinstate such feelings'. In the introduction: 'Thus, emotional habituation to both negative and positive feelings may lead to increased risk-taking'. Given the current dataset, I am trying to imagine how the authors could test this - the first thing that comes to mind is to use the baseline emotion (before the experiment started) and other trait measures and see if they predict risk-taking, which the authors have done and found no relationship. Therefore, I would strongly suggest the authors tone down their language.**
 - We agree that the results should be presented and interpreted with utmost caution as not to elicit the sense of a causal conclusion. We have now altered any wording that could be viewed as a casual interpretation of our result (pp. 3-4, 14-16). In the introduction, however, we introduce the theoretical reason why a relationship between emotional habituation and risk-taking escalation would be hypothesized. It is critical for the reader to understand the theory and so we keep that language while making it crystal clear that we are only testing a correlation.
- 2- Another thing that I am confused about in the paper is the fact that people keep taking risks and that their anxiety seems to be reducing. If as the authors have argued a lower level of positive/negative emotion promotes more risk taking &**

if we assume people have an optimal level of anxiety/excitement, would they eventually converge to a stable level of emotion but taking more and more risk (since emotion itself will get habituate as time goes by?) Do the authors agree with the above argument but just the current task is not long enough for them to converge to that saturation point?

- If we understand the question correctly, the reviewer is asking whether emotional responses would continue to habituate over time, leading to a continuous escalation of risk-taking, and whether participants might eventually converge to a stable emotional state if the task were long enough. Theoretically speaking, as time progresses, repeated exposure to risk can indeed lead to emotional habituation. This may result in a steady increase in risk-taking. This process would likely continue uninterrupted until one of two conditions occurs: either a change is introduced (e.g., an external cue or intervention), or a catastrophic outcome is encountered. When such an outcome occurs, it may trigger a **dishabituation effect**, whereby emotional arousal are reinstated and risk-taking escalation is halted (Sharot & Sunstein, 2024).

3- If the authors ditched the hint at causality, then I am not sure that this paper has added too much to what we already know (given the current dataset), since as the authors have mentioned, emotion habituates in response to an emotion-evoking stimulus is an established finding. Extra experiments can potentially make the current study interesting e.g., as the authors have alluded to, manipulate emotion separately on a trial-by-trial basis (so that the authors can manipulate the rate of the outcome of emotion habituation) and see if it causally influences risk taking behavior. Another way could be using a yoked design and making limitations/requirements on the amount of risk the participant needs to take, and comparing the emotion habituation level between (1) voluntarily taking however amount of risk and (2) pre-specified high/low risk level.

- We thank the reviewer for providing us with the opportunity to detail the novelty and importance of the work. While emotional habituation is well established, its impact on decision-making is not. In fact, an extensive review of the literature (Sharot & Sunstein, 2024) has identified only one other study directly measuring the relationship of emotional habituation to decision-making. That other study (Garrett et al., 2016) related emotional habituation to dishonesty behavior, not to risk taking. This behavioral link is not trivial. In many decision-making frameworks, risk preference is treated as a stable trait or as influenced by incidental emotions. In contrast, our study suggests that within-task emotional habituation may explain individual differences when facing similar objective outcomes. This study does not only shine new light on risk taking behavior - it is also likely to inspire other studies linking emotional habituation to different forms of behavior. Moreover, the study's methodology is likely to have a large impact. It demonstrates how VR can be used to study questions related to emotions, risk and

decision making that were very difficult to examine before due to ethical and practical limitations. While manipulating emotional habituation is beyond the scope of this current study in its present form, this study provides an important empirical foundation upon which a range of future investigation may build. We now add the above to the Discussion section (p. 15).

Other comments:

4- Though the authors have argued for the link between the in-lab measure and real-life counterpart (afraid of heights), I am not sure that this can be directly linked to risk escalation in economic domains - is there any work suggesting that physical risk-taking correlates with financial risk taking? The author should discuss this point in the discussion.

➤ We thank the reviewer for suggesting this. Some studies suggest that risk-taking tendencies are domain general. For example, individuals who seek out physical risks (e.g., extreme sports or health-related risks) often show a greater tendency toward financial risk-taking (Nicholson et al., 2005; Rolison et al., 2014; Reeck et al., 2022). Emotional habituation may contribute to this cross-domain tendency. One challenge in studying financial risk escalation in the laboratory is the relatively low emotional engagement elicited by small-stakes that are usually on offer. Future studies could address this by using naturalistic field experiments. We now mention this in the Discussion section (pp. 15-16).

5- In the introduction, the authors argue that 'while strong fear and anxiety will curb risk-taking, the absence of such affective signals will free people to take risks' - this is not necessarily true. In FeldmanHall et al. (2016), they found that this depends on whether the situation is risk or ambiguous. Under ambiguity, higher physiological arousal predicts more ambiguous seeking behavior. The concept of risk in their paper does not necessarily map onto the current paper - if I have to choose, I would say that risk taking in the current experiment is more related to ambiguity than risk since the probability of falling down is unknown to the participant. Relatedly, RE the relationship between physiological arousal and risk-taking, this seems can be tested by some physiological data, which the authors seem to have, at least for a subset of all the participants (i.e., GSR data for the first 53 participants). I am not sure if the data will be underpowered to test this, but it would be interesting to see this set of results.

➤ We thank the reviewer for highlighting this paper which we now cite in the introduction (p. 3). Empirically, in the current experiment it appears that a decrease in anxiety is related to more risk taking. Following the reviewer's question, we now analyze the Galvanic Skin Response (GSR). We found that GSR response exhibits habituation, though at a trend level due to low N ($\beta = -0.01$, $t(44.99) = -1.89$, $P = 0.064$). While numerically across individuals the greater the

rate of GSR habituation the greater the risk escalation, this effect did not reach significance ($\beta = -0.19$, $t(43) = -1.60$, $P = 0.11$). This null result is likely due to insufficient statistical power; a Bayes Factor ($BF_{10} = 0.82$) indicated inconclusive evidence, and power analysis suggests a sample of $N = 212$ would be needed to detect such an effect reliably. These data hint at the possibility that arousal habituation is associated with risk-taking escalation. We do not add these analyses in the supplement because of the low N but can do so if the editor deems it appropriate.

6- In the methods, the authors mentioned that if the participant fell off, it is calculated as the maximum length - why is this the case? what is the proportion of the data that belongs to falling off?

- Just under half of the participants chose to step off the plank at least once during the task. And on average did so just under two times (1.89). We gave these trials the same score as walking to the edge as it is the most 'risky' behavior. However, to ensure that this coding does not impact on the results we reanalyzed the data excluding all trials in which participants stepped off the plank and find the results unchanged (see Supplementary Results pp. 1-2).

7- Is there any correlation between anxiety and excitement? Do the authors believe that anxiety and excitement have a common cause, or do they originate separately?

- There was a small correlation between the average ratings of anxiety and excitement across subjects ($\beta = 0.158$, $t(157) = 2.01$, $P = 0.046$). That is, subjects who on average gave higher/lower anxiety ratings were slightly more likely to provide higher/lower excitement ratings on average. Moreover, predicting anxiety on a trial-by-trial level from excitement in a mixed linear model with random intercepts and slopes also revealed an association between the two ($\beta = 0.39$, $t(135.208) = 8.98$, $P < 0.001$). This suggests that both may be partially related to a common feature such as arousal habituation. These additional analyses have now been included in the Results section (p. 7).

8- The second paragraph mentions Figure 1c as the graph showing some coefficients - I am not sure how a graphical illustration of the experiment suffices this goal - maybe the authors mean a different graph?

- We thank the Reviewer for noting this oversight. We have corrected the reference to Figure 1d in the manuscript (pp. 7-8).

References

- Beck, A. T., Emery, G., & Greenberg, R. L. (2005). *Anxiety disorders and phobias: A cognitive perspective*. Basic Books/Hachette Book Group.
- Biedermann, S. V., Biedermann, D. G., Wenzlaff, F., Kurjak, T., Nouri, S., Auer, M. K., Wiedemann, K., Briken, P., Haaker, J., Lonsdorf, T. B., & Fuss, J. (2017). An elevated plus-maze in mixed reality for studying human anxiety-related behavior. *BMC Biology*, *15*(1), 125. <https://doi.org/10.1186/s12915-017-0463-6>
- Bieling, P. J., Antony, M. M., & Swinson, R. P. (1998). The State-Trait Anxiety Inventory, Trait version: structure and content re-examined. *Behaviour Research and Therapy*, *36*(7–8), 777–788. [https://doi.org/10.1016/S0005-7967\(98\)00023-0](https://doi.org/10.1016/S0005-7967(98)00023-0)
- Brooks, A. W. (2014). Get excited: Reappraising pre-performance anxiety as excitement. *Journal of Experimental Psychology: General*, *143*(3), 1144–1158.
- Daniel-Watanabe, L., & Fletcher, P. C. (2021). Are Fear and Anxiety Truly Distinct? *Biological Psychiatry Global Open Science*, *2*(4), 341–349. <https://doi.org/10.1016/j.bpsgos.2021.09.006>
- Davis, M., Walker, D. L., Miles, L., & Grillon, C. (2010). Phasic vs sustained fear in rats and humans: role of the extended amygdala in fear vs anxiety. *Neuropsychopharmacology*, *35*(1), 105–135. <https://doi.org/10.1038/npp.2009.109>
- FeldmanHall, O., Glimcher, P. W., Baker, A., & Phelps, E. A. (2016). Emotion and decision-making under uncertainty: Physiological arousal predicts increased gambling during ambiguity but not risk. *Journal of Experimental Psychology: General*, *145*(10), 1255–1262.
- Garrett, N., Lazzaro, S.C., Ariely, D., & Sharot, T. (2016). The Brain Adapts to Dishonesty. *Nature neuroscience*, *19*, 1727 - 1732.
- Gromer, D., Madeira, O., Gast, P., Nehfischer, M., Jost, M., Müller, M., Mühlberger, A., & Pauli, P. (2018). Height simulation in a virtual reality CAVE system: Validity of fear responses and effects of an immersion manipulation. *Frontiers in Human Neuroscience*, *12*, 372. <https://doi.org/10.3389/fnhum.2018.00372>
- Gromer, D., Reinke, M., Christner, I., & Pauli, P. (2019). Causal interactive links between presence and fear in virtual reality height exposure. *Frontiers in Psychology*, *10*, 141. <https://doi.org/10.3389/fpsyg.2019.00141>
- Grupe, D. W., & Nitschke, J. B. (2013). Uncertainty and anticipation in anxiety: An integrated neurobiological and psychological perspective. *Nature Reviews Neuroscience*, *14*(7), 488–501. <https://doi.org/10.1038/nrn3524>
- Gray, J. A. (1991). Fear, panic, and anxiety: What's in a name? *Psychological Inquiry*, *2*(1), 77–78. https://doi.org/10.1207/s15327965pli0201_18

Ketonen, E. E., Salonen, V., Lonka, K., & Salmela-Aro, K. (2023). Can you feel the excitement? Physiological correlates of students' self-reported emotions. *The British Journal of Educational Psychology*, Article e12534.

Knutson, B., Wimmer, G. E., Kuhnen, C. M., & Winkielman, P. (2008). Nucleus accumbens activation mediates reward cues' influence on financial risk-taking. *NeuroReport*, *19*, 509–513.

Kuhnen, C. M., & Knutson, B. (2011). The influence of affect on beliefs, preferences, and financial decisions. *Journal of Financial and Quantitative Analysis*, *46*, 605–626.
<https://doi.org/10.1017/S0022109011000123>

Kuppens, P., Tuerlinckx, F., Russell, J. A., & Barrett, L. F. (2013). The relation between valence and arousal in subjective experience. *Psychological Bulletin*, *139*(4), 917–940.
<https://doi.org/10.1037/a0030811>

Lang, P. J. (1995). The emotion probe: Studies of motivation and attention. *The American Psychologist*, *50*(5), 372–385. <https://doi.org/10.1037/0003-066X.50.5.372>

Miu, A. C., Heilman, R. M., & Houser, D. (2008). Anxiety impairs decision-making: Psychophysiological evidence from an Iowa Gambling Task. *Biological Psychology*, *77*, 353–358.

Nicholson, N., Soane, E., Fenton-O'Creevy, M., & Willman, P. (2005). Personality and domain-specific risk taking. *Journal of Risk Research*, *8*(2), 157–176.
<https://doi.org/10.1080/1366987032000123856>

Reeck, C., Mullette-Gillman, O. A., McLaurin, R. E., & Huettel, S. A. (2022). Beyond money: Risk preferences across both economic and non-economic contexts predict financial decisions. *PloS one*, *17*(12), e0279125. <https://doi.org/10.1371/journal.pone.0279125>

Rolison, J. J., Hanoch, Y., Wood, S., & Liu, P. J. (2014). Risk-taking differences across the adult life span: a question of age and domain. *The journals of gerontology. Series B, Psychological sciences and social sciences*, *69*(6), 870–880.
<https://doi.org/10.1093/geronb/gbt081>

Russell, J. A., & Barrett, L. F. (1999). Core affect, prototypical emotional episodes, and other things called emotion: Dissecting the elephant. *Journal of Personality and Social Psychology*, *76*(5), 805–819. <https://doi.org/10.1037/0022-3514.76.5.805>

Sharot, T., & Sunstein, C. R. (2024). *Look again: The power of noticing what was always there*. Little, Brown Book Group.